# Development and evaluation of PlasmoPod: A cartridge-based nucleic acid amplification test for rapid malaria diagnosis and surveillance

Philippe Bechtold[1,2]*, Philipp Wagner[3,4], Salome Hosch[3,4], Michele Gregorini[1,2], Wendelin J. Stark[1,2], Jean Chrysostome Gody[5], Edwige Régina Kodia-Lenguetama[6], Marilou Sonia Pagonendji[7], Olivier Tresor Donfack[8], Wonder P. Phiri[8], Guillermo A. García[8], Christian Nsanzanbana[3,4], Claudia A. Daubenberger[3,4], Tobias Schindler[3,4☯]*, Ulrich Vickos[9,10☯]

1 Institute for Chemical and Bioengineering, ETH Zurich, Zuerich, Switzerland, 2 Diaxxo AG, Zuerich, Switzerland, 3 Swiss Tropical and Public Health Institute, Basel, Switzerland, 4 University of Basel, Basel, Switzerland, 5 Paediatric Hospital and University Complex of Bangui, Bangui, Central African Republic, 6 National Blood Transfusion Centre, Ministry of Health, Bangui, Central African Republic, 7 Laboratory of Parasitology, Institute Pasteur of Bangui, Bangui, Central African Republic, 8 MCD Global Health, Malabo, Equatorial Guinea, 9 Infectious and Tropical Diseases Unit, Department of Medicine, Amitié Hospital, Bangui, Central African Republic, 10 Microbiology and Diagnostic Immunology Unit, Bambino Gesù Children's Hospital, IRCCS, Rome, Italy

☯ These authors contributed equally to this work.
* p.bechtold@diaxxo.com (PB); tobias.schindler@swisstph.ch (TS)

**Data Availability Statement:** All data that support the findings of this study are available as a

## Abstract

Malaria surveillance is hampered by the widespread use of diagnostic tests with low sensitivity. Adequate molecular malaria diagnostics are often only available in centralized laboratories. PlasmoPod is a novel cartridge-based nucleic acid amplification test for rapid, sensitive, and quantitative detection of malaria parasites. PlasmoPod is based on reverse-transcription quantitative polymerase chain reaction (RT-qPCR) of the highly abundant *Plasmodium* spp. 18S ribosomal RNA/DNA biomarker and is run on a portable qPCR instrument which allows diagnosis in less than 30 minutes. Our analytical performance evaluation indicates that a limit-of-detection as low as 0.02 parasites/µL can be achieved and no cross-reactivity with other pathogens common in malaria endemic regions was observed. In a cohort of 102 asymptomatic individuals from Bioko Island with low malaria parasite densities, PlasmoPod accurately detected 83 cases, resulting in an overall detection rate of 81.4%. Notably, there was a strong correlation between the Cq values obtained from the reference RT-qPCR assay and those obtained from PlasmoPod. In an independent cohort, using dried blood spots from malaria symptomatic children living in the Central African Republic, we demonstrated that PlasmoPod outperforms malaria rapid diagnostic tests based on the PfHRP2 and panLDH antigens as well as thick blood smear microscopy. Our data suggest that this 30-minute sample-to-result RT-qPCR procedure is likely to achieve a diagnostic performance comparable to a standard laboratory-based RT-qPCR setup. We believe that the PlasmoPod rapid NAAT could enable widespread accessibility of high-quality and cost-effective molecular malaria surveillance data through decentralization of testing and surveillance activities, especially in elimination settings.

supplementary document uploaded to the journal's website.

**Funding:** Funding for PB, MG and WJS was provided by the Botnar Research Centre for Child Health as part of the Fast Track Call for Acute Global Health Challenges as well as the BRIDGE programme by Swiss National Science Foundation and Innosuisse. TS is supported by the Bioko Island Malaria Elimination Project (BIMEP). BIMEP is a public private partnership funded through the Government of Equatorial Guinea, Marathon Oil, Noble Energy, SonaGas, GEPetrol, and Atlantic Methanol (AMPCO). The funders had no role in study design, data collection and analysis, decision to publish, or preparation of the manuscript.

**Competing interests:** We have read the journal's policy and the authors of this manuscript have the following competing interests: PB, MG, and WJS are co-inventors on a corresponding patent application around the diax-xoPCR platform. MG, TS and WJS are shareholders of the ETH spin-off company Diaxxo AG. All other authors declare no competing interests.

## Introduction

Malaria is an infectious disease caused by different *Plasmodium* spp. species and is transmitted through female *Anopheles* spp. mosquitoes to humans [1]. Although significant progress on combating the spread of the disease has been achieved, more than 600'000 people still die annually [2]. Considerable improvements in the accuracy and availability of diagnostics need to be achieved to reduce the overall burden of the disease further with the aim of malaria elimination [3]. Currently employed methods comprise blood smear microscopy, antigen rapid diagnostic tests (RDT) and nucleic acid amplification tests (NAAT) [4]. Diagnosis of malaria by light microscopy using Giemsa-stained thick or thin blood smears has been the gold standard since the early $20^{th}$ century. Well trained and experienced microscopists can reach a limit of detection (LOD) of 50–100 parasites/µL blood [5]. Expert microscopists are however chronically lacking, and the sample throughput is rather low. RDTs are based on detection of parasite antigens in blood and can provide valuable and rapid answers in remote areas without the need for extensive training. They are low-cost and widely available with 419 million WHO prequalified malaria RDTs sold globally in 2020 [2]. Despite the improvements in the quality of malaria RDTs through programs like the WHO product testing [6, 7], their sensitivity remains limited as they detect antigens without involving target molecule amplification. A LOD, of 100–200 parasites/µL for PfHRP2-based RDTs [8] and about 1000 parasites/µL for panLDH-based RDTs [9] renders them unsuitable for surveillance in endemic and low-transmission environments and in elimination settings [10]. The presence of *P. falciparum* strains with *pfhrp2* gene deletions poses an additional challenge for malaria surveillance as these strains cannot be detected by PfHRP2-based RDTs [11]. NAAT such as reverse transcription quantitative polymerase chain reaction (RT-qPCR) have shown a LOD of 0.05 parasites/µL [12], which is by a factor of 1000 more sensitive than microscopy. This approach is especially favourable in areas where a high proportion of asymptomatic malaria carriers are living, maintaining the transmission cycle of the parasite [13, 14]. However, in resource-constraint settings the use of highly sensitive RT-qPCR based malaria testing has been restricted to well-equipped centralized laboratoriesdue to high initial investment cost, sophisticated supply chain management and shortage of trained laboratory personnel [15]. Simple, rapid, highly sensitive and reliable molecular diagnostic tools are needed more than ever in conjunction with a functional surveillance system to enable and sustain malaria elimination.

The diaxxoPCR technology is based on a portable and easy-to-use qPCR instrument, which is designed for rapid identification, quantification and genotyping of pathogens at affordable costs. Only minimal hands-on-operations are needed, and pathogens can be detected in less than 30 minutes, based on an innovative temperature control strategy that allows reaching unprecedented high heating and cooling rates ($> 13°C/s$) during the PCR [16]. Importantly, no cold-chain during shipping, storage or usage of the reagents is needed since the RT-qPCR reactions are run in aluminium-based cartridges, which come preloaded with all reagents in dried form. The cartridges are equipped with a total of 20 wells that can be loaded according to specific requirements. Each well accommodates a single qPCR assay to measure one sample. With a single qPCR assay, the maximum number of patient samples per cartridge is 20. However, when incorporating controls or a standard curve, the number of samples per cartridge will be reduced accordingly.

For mobile testing applications, the device can be powered using a car battery and the results can be accessed directly on the device's screen or through the browser of a smartphone or laptop. The diaxxoPCR device performs RT-qPCR amplification with very small reagent and sample input volumes, rendering it highly cost efficient in addition to its unparalleled speed. The PlasmoPod offers cost advantages over standard RT-qPCR assays due to its minimal reagent volumes, with an estimated cost-per-sample of around EUR 1.5 in small-size

batches and the potential to decrease below EUR 1.0 at scale. The diaxxoPCR platform has delivered results comparable to state-of-the-art qPCR devices for SARS-CoV-2 detection and genotyping [17]. The instrument achieved excellent diagnostic performance when tested with RNA extracted from culture-derived SARS-CoV-2 Variants of Concern (VOC) lineages and clinical samples collected in Equatorial Guinea, Central-West Africa [17].

In the current study, we describe the development of a *Plasmodium* spp. cartridge for the diaxxoPCR device referred to as "PlasmoPod". We further report on the performance of PlasmoPod as a rapid and highly sensitive NAAT-based diagnostic tool for malaria and compare it to other currently available diagnostic tests using samples collected from children and adults living in two different Central African countries.

## Materials and methods

### Ethics statement

The malaria indicator survey conducted on Bioko Island, Equatorial Guinea was approved by the Ministry of Health and Social Welfare of Equatorial Guinea and the Ethics Committee of the London School of Hygiene & Tropical Medicine (Ref. No. LSHTM: 5556). Written informed consent was obtained from all adults and from parents or guardians of children who agreed to participate. Only samples for which an additional consent for molecular analysis was obtained were included in this study. The study in Bangui, Central African Republic was conducted in accordance with the Declaration of Helsinki and was approved by the Ethics and Scientific Committee from the University of Bangui (approval n˚3/UB/FACSS/CSCVPER/PER) and by the Ministry of Health of the Central African Republic (approval n˚0277/MSPP/CAB/DGSPP/DMPM/ SMEE du 05 août 2002) as part of the communicable and endemic diseases surveillance diagnostic program. The patients were informed about the objectives of the study and nature of their participation. Then, written and signed informed consents were obtained from the participants or the parents on behalf of their children.

### PlasmoPod NAAT development and analytical performance evaluation

The experiments on the diaxxoPCR platform were performed using the PlasmoPod cartridges supplied by Diaxxo AG (Zuerich, Switzerland). The 20 well cartridges contain all reagents necessary for running a RT-qPCR in preloaded and in dried form. The *Plasmodium* spp. assay used for PlasmoPod is a TaqMan probe-based qPCR assay which uses a 6-Carboxyfluorescein (6-FAM) labelled probe to enable the detection of a specific amplification product produced during PCR. Published oligonucleotide sequences and concentrations for detection of *Plasmodium* spp. parasites are used [18]. Analytical performance of PlasmoPod was evaluated with purified NAs from different relevant pathogens. Cross-reactivity was tested against DNA or RNA extracted from bacteria (*Salmonella enterica subsp. enterica* serovar Typhi), viruses (Dengue virus serotype 3, Chikungunya virus, Yellow fever virus and Zika virus) and closely related apicomplexan parasites (*Cryptosporidium parvum* and *Cryptosporidium hominis*). Additionally, the assay's specificity was evaluated using *Plasmodium* spp.-free human blood and serum samples from four different donors. The sensitivity, accuracy, and reproducibility of the PlasmoPod NAAT was assessed using DNA extracted from cultivated and synchronized ring-stage NF54 *P. falciparum* parasites. Four technical replicates from a total of 15 DNA titration steps with concentrations ranging from 500 to 0.0008 parasites/μL, were analysed with PlasmoPod using the following cycling parameters on the diaxxoPCR instrument: reverse transcription of 300 seconds at 53˚C, initial polymerase activation for 60 seconds at 90˚C and then 45 cycles of 10 seconds at 94˚C and 20 seconds at 56˚C. Raw data was analysed by diaxxoPCR software and Cq values were automatically assigned to the samples.

## Collection and characterization of samples from the asymptomatic malaria cohort

Bio-banked samples collected during The Malaria Indicator Surveys (MIS) conductedSurvey on Bioko Island, Equatorial Guinea in 2018 and 2019 were used for this study. The MIS involved voluntary participation of permanent residents and short-term visitors. The volunteers included in the survey were classified as asymptomatic for malaria and the survey was conducted at their respective places of residence. They were tested for malaria using the CareStart Malaria HRP2/pLDH Combo RDT. The used RDTs were stored at room temperature in plastic bags with desiccants and subsequently transported to the Swiss Tropical and Public Health Institute for additional molecular analysis. Total nucleic acids were extracted by the "Extraction of Nucleic Acids from RDTs" (ENAR) protocol and [19, 20] and the pan-*Plasmodium* spp. 18S ribosomal DNA and RNA molecules were targeted [18, 21] and detected by a highly-sensitive RT-qPCR (herein referred to Pspp18S RT-qPCR assay) [19]. A total of 102 samples, found positive by the Pspp18S RT-qPCR assay (Quantification Cycle (Cq)Cq values <40),) were included into the PlasmoPod evaluation study.

## Collection and characterization of samples from the clinical malaria cohort

The samples from the clinical cohort were collected at the Paediatric Hospital and University Complex of Bangui (CHUPB), located in the Central African Republic (CAR). Collection took place between March 8th and 13th, 2021. The patients were children aged between 2 months and 15 years that were admitted to the emergency department with fever as their main clinical symptom. In case malaria was suspected and after obtaining informed consent from their legal guardians, whole blood samples were collected in EDTA blood collection tubes. A malaria RDT (A&B Rapid Test Malaria P.f./Pan, Luca, Italy)), thick blood smear (TBS) microscopy and a complete blood count were routinely performed. An aliquot of the whole blood was prepared as dried blood spots (DBS) on filter papers. The DBS were stored at room temperature and sent to the Swiss Tropical and Public Health Institute, Basel, Switzerland for further molecular analysis.

## Molecular characterization of malaria parasites identified among clinical cohort samples with reference molecular assays

A molecular reference dataset from the DBS collected in the CAR was established to be used as a gold standard against which the performance of PlasmoPod was compared to. The reference dataset included the species identification of *Plasmodium* spp. positive samples as well as the analysis of the *pfhrp2/3* deletion status and quantification of the parasite density of all *P. falciparum* positive samples. Briefly, the New Extraction Method (NEM) protocol developed by Zainabadi *et al.* was used to extract total nucleic acids (NA), including DNA and RNA, from the DBS [22]. In short, one entire DBS, which corresponds to 30–50 μL of whole blood, was lysed at 60˚C for 2 h. NAs were subsequently purified and eluted in 100 μL elution buffer as described elsewhere [19]. The same Pspp18S RT-qPCR assay targeting pan-*Plasmodium* spp. 18S ribosomal DNA and RNA molecules as for the analysis of the asymptomatic malaria cohort, was used. The Pspp18S assay was analysed and samples with Cq values <40 were considered malaria positive. All samples positive for the Pspp18S assay were analysed by species-specific qPCR assays as described previously [23]. All samples positive for *P. falciparum* were screened for *pfhrp2* and/or *pfhrp3* deletions using a multiplex qPCR assay detecting *pfhrp2/3* deletions [24]. Only samples with a Cq value < 35 for the internal control of the *pfhrp2/3* deletion assay were considered eligible for analysis of deletion status. All reference qPCR and RT-qPCR assays were run on a Bio-Rad CFX96 Real-Time PCR System (Bio-Rad Laboratories,

California, USA). Samples were analysed in duplicate with positive (DNA from *P. falciparum* strain NF54) and non-template controls (molecular biology grade $H_2O$) added to each run.

## qPCR-based quantification of *P. falciparum* parasite density of clinical malaria samples

The *P. falciparum* parasite density was determined based on the amplification and detection of the *P. falciparum*-specific single copy gene ribonucleotide reductase R2_e2 (herein referred to as PfRNR2 assay) [24, 25]. Briefly, the WHO International Standard for *Plasmodium falciparum* DNA for NAAT-based assays (*PfIS*) [26] was used to generate a serial dilution in parasite-free whole blood, ranging from 0.01 to 100'000 parasites/μL. Thirty μL of each dilution step was put on a DBS and dried, followed by NA extraction and qPCR quantification by the PfRNR2 assay. The resulting standard curve, including the slope and y-axis intercept, was used to quantify the parasite densities in the clinical samples. Based on a cut-off value of 5000 parasites/μL, the malaria positive children were categorized into high and moderate parasite density infection groups. The parasite density cut-off was determined based on clinical study criteria for symptomatic and severe malaria [27–29]. In case the PfRNR2 assay was negative while the more sensitive Pspp18S assay was positive the child was assigned to the moderate parasite density group.

## PlasmoPod NAAT evaluation using the asymptomatic and clinical malaria cohort samples

**Asymptomatic malaria cohort.** One replicate of 4.5 μL of extracted total NAs was loaded into a well of the PlasmoPod and run on the diaxxoPCR device using its standard cycling program and data analysis as described above. Raw data was analysed by diaxxoPCR software and Cq values were automatically assigned to the samples.

**Clinical malaria cohort.** Rapid extraction was performed on a single DBS punch with a 3mm diameter. The DBS punch was submerged in 100 μL of a 5% Chelex (Bio-Rad, California, USA) solution and heated to 95˚C for 3 min. The supernatant of the resulting solution was used directly for PlasmoPod analysis. Briefly, 4.5 μL of eluate per sample were loaded in duplicates onto the 20-well cartridge, covered with paraffin oil (Sigma-Aldrich, St. Louis, USA) and cycled for a duration of 25 minutes (for 45 cycles) in the diaxxoPCR device. Each run contained two wells with positive (DNA from *P. falciparum* strain NF54) and two wells of a non-template control (molecular biology grade $H_2O$) control. The cycling parameters on the diaxxoPCR were as follows: reverse transcription of 300 seconds at 50˚C, initial polymerase activation for 60 seconds at 92˚C and then 45 cycles of 2 seconds at 92˚C and 15 seconds at 55˚C. Raw data was analysed by diaxxoPCR software and Cq values were automatically assigned to the samples. Specimens with amplification with a Cq < 40 in 2/2 of replicates were considered positive.

## Data analysis

Statistical analysis and data visualization was performed using the R statistical language (version 4.1.2) based on packages dplyr, epiR, ggplot2, ggpubr, gridExtra, readxl, reshape2, scales, tidyr, tidyverse, cowplot, and plyr.

## Results

### PlasmoPod is a cartridge-based NAAT for rapid *Plasmodium* spp. detection

The diaxxoPCR device is a novel, small-scale and standalone qPCR instrument (Fig 1A) which can be used to run and analyse ready-to-use cartridges which contains all RT-qPCR reagents

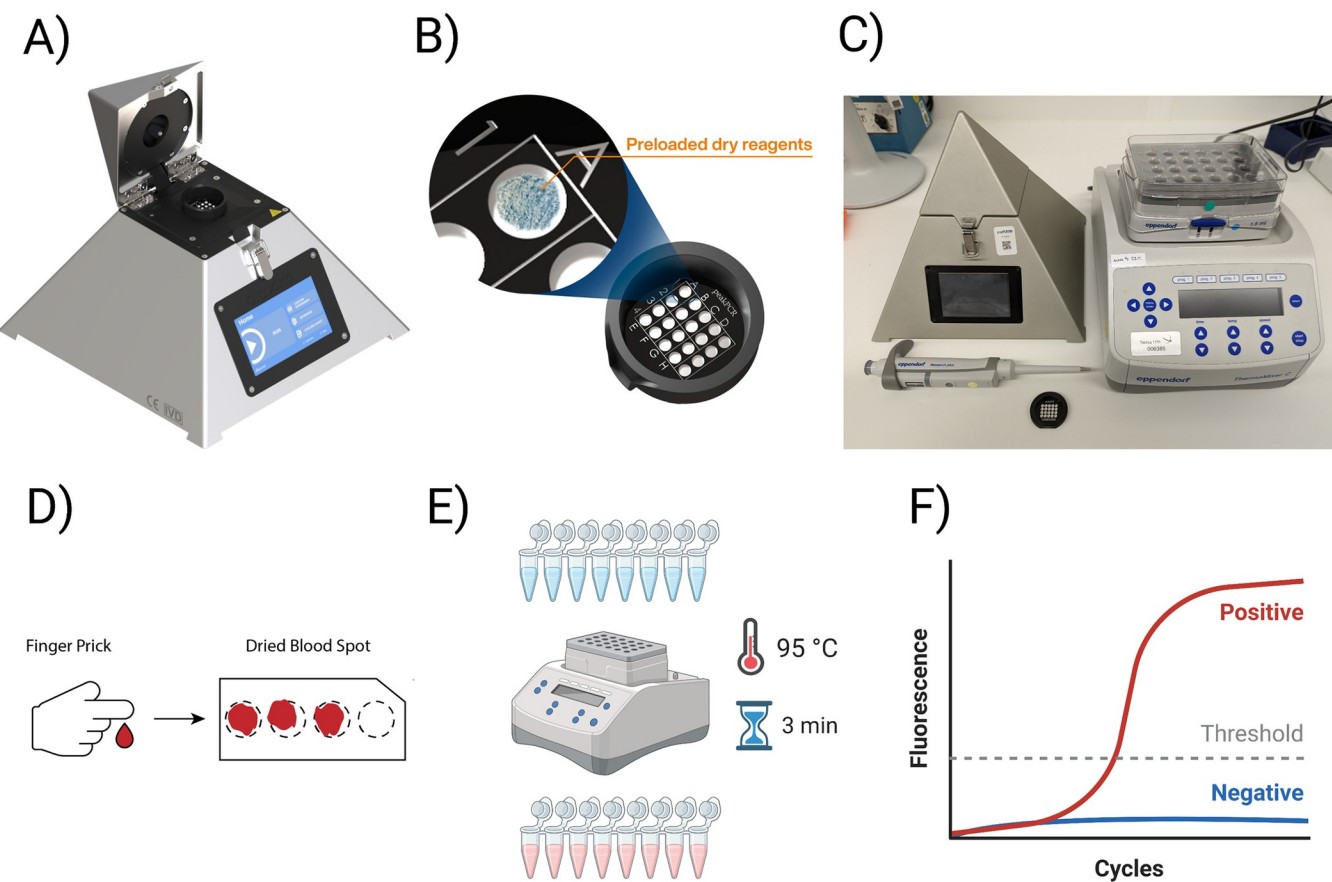

**Fig 1. DiaxxoPCR and PlasmoPod setup for rapid molecular malaria detection.** A) The DiaxxoPCR instrument is a pyramid-shaped stand-alone device for rapid qPCR cycling and fluorescence acquisition. (B) The PlasmoPod assay is based on a single-use cartridge which is pre-loaded with all qPCR reagents in dried form. (C) The PlasmoPod laboratory setup consisting of the diaxxoPCR device, a heatblock, a pipette and PlasmoPod cartridges. (D) Finger prick blood is sampled and stored as dried blood spots (DBS). (E) The setup for rapid NA extraction from DBS. (F) The NA are amplified and signals analysed using the diaxxoPCR device and its integrated analysis software. Images A and B are republished from diaxxo AG under a CC BY license, with permission from Dr Michele Gregorini (who is an author of this manuscript), original copyright [diaxxo AG, 2020–2022]. The granted permission to the manuscript has been uploaded. Partially created with Biorender.com.

and oligonucleotidesoligo nucleotides in dried form (Fig 1B). The cartridge developed and evaluated in this study was named "PlasmoPod". The diaxxoPCR device was selected as the core component for our minimal laboratory setup for molecular malaria diagnosis due to its compact size and robustness (Fig 1C). Unlike many other devices, the diaxxoPCR does not incorporate moving parts, making it highly suitable for mobile testing applications. In terms of dimensions, the device is comparable to a standard laboratory heating block, ensuring ease of portability and integration into various testing environments. As a first application for malaria diagnosis using diaxxoPCR, we developed a novel molecular diagnostic approach for rapid, sensitive, and quantitative detection of malaria parasites from blood sampled and stored on DBS. This approach included a rapid NA extraction procedure from DBS by submerging and boiling a single 3 mm diameter DBS punch in a Chelex solution (Fig 1D). During the 3-minute incubation step the blood preserved on the DBS is dissolved into the solution, the cells are lysed, and potential PCR inhibitors are removed by the Chelex (Fig 1E). Without any further processing 4.5 µL of this solution are directly loaded into a well of the PlasmoPod. Using the diaxxoPCR device, the reverse transcription and a total of 45 PCR cycles are run in less than 30 minutes. The results can then be accessed through the screen of the diaxxoPCR device, a

connected smartphone or a computer (Fig 1F). In the current study we evaluated the Plasmo-Pod cartridge run on the diaxxoPCR rapid PCR device by comparison with a standard laboratory-based diagnostic approach for malaria based on state-of-the-art NA extraction procedure and RT-qPCR detection.

## The analytical performance evaluation of PlasmoPod enables quantitative detection of *Plasmodium* spp. parasites with high specificity, reproducibility and sensitivity

The primary objective of the analytical performance evaluation was to assess the performance of the qPCR itself. Thus, we utilized purified NAs, without considering the rapid extraction procedure's potential impact on the analytical performance. The potential for cross-reactivity and unspecific amplification of PlasmoPod was tested with purified NA from a range of pathogens co-circulating in malaria endemic countries and parasite-free human-derived samples (**Fig 2A**). PlasmoPod measurements with bacteria (*Salmonella enterica subsp. enterica* serovar Typhi), viruses (Dengue virus serotype 3, Chikungunya virus, Yellow fever virus and Zika virus), closely related apicomplexan parasites (*Cryptosporidium parvum* and *Cryptosporidium hominis*) and *Plasmodium* spp.-free human blood resulted in delta fluorescence values (endpoint minus baseline fluorescence) and maximum amplification curve slope values below the pre-defined positivity cut-off values. As comparison, results obtained with purified total NAs from culture-derived ring-stage synchronized *P. falciparum* parasites analysed at different concentrations are shown. To demonstrate the ability to also detect non-*falciparum* human pathogenic *Plasmodium* spp. species, we analysed clinical samples positive for *P. vivax*, *P. ovale* spp. and *P. malariae*. For *P. falciparum*, the two dilutions with the lowest input concentration were

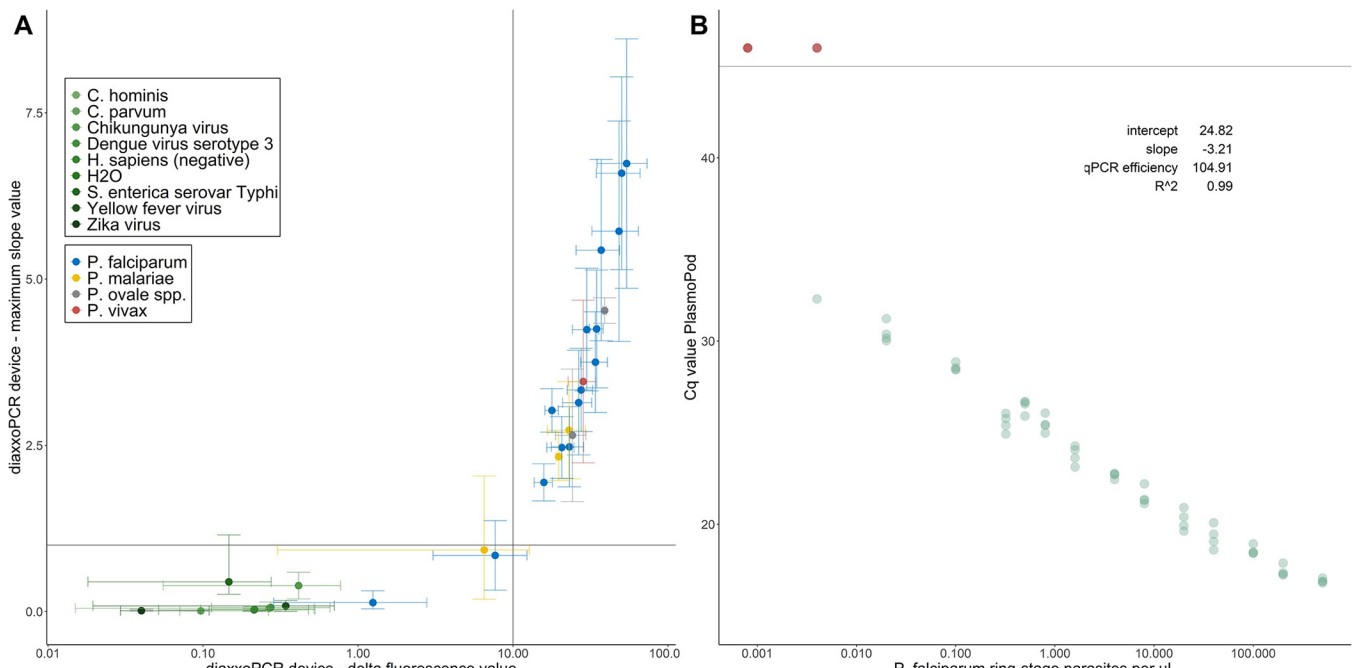

**Fig 2. Analytical performance evaluation of PlasmoPod.** (A) DiaxxoPCR derived delta fluorescence and maximum slope after amplification with nucleic acid from different pathogens. Lines indicate the delta fluorescence positivity cutoff (vertical) and maximum slope positivity cutoff (horizontal). (B) Cq value of PlasmoPod versus dilutions of NAs extracted from culture-derived ring-stage synchronized *P. falciparum* parasites. Sample without amplification are colored in red.

below the pre-defined positivity thresholds but still distinguishable from the non-malaria samples (**Fig 2A**). Using the same serial dilution of DNA extracted from culture-derived ring-stage synchronized *P. falciparum* parasites, the relationship between concentration and Cq values was established (**Fig 2B**). Four replicates of all concentrations were run on three different PlasmoPod cartridges. The lowest concentration which resulted in a positive signal for all four replicates was 0.02 parasites/μL. The relationship between Cq values and parasite concentration is described by an $R^2$ value of 0.99. The slope of -3.21 translates into an almost perfect qPCR efficiency of 104.91%.

## Diagnostic performance of PlasmoPod evaluated with samples from asymptomatic parasite carriers using NAs extracted from archived RDTs

To assess the performance of PlasmoPod as a sensitive molecular tool for malaria surveillance, we examined extracted NA from 102 asymptomatic individuals carrying malaria parasites, who were part of the annual malaria indicator survey conducted on Bioko Island, Equatorial Guinea. For this study, only samples found positive by a previous qPCR screening were included. Among the participants, 47 tested positive for both PfHRP2 and panLDH antigens, 23 for panLDH alone, 12 for PfHRP2 alone, and 20 were negative for both antigens. NAs were extracted directly from the blood stored on the archived RDTs and analyzed using the laboratory RT-qPCR assay targeting Plasmodium spp. 18S rDNA/rRNA as the gold standard diagnostic test. All 102 individuals had detectable *Plasmodium* spp. NA on their archived RDTs, with Cq values ranging from 23.6 to 39.0 and a median of 33.4. PlasmoPod correctly identified 83 out of the 102 samples, yielding an overall detection rate of 81.4%. A strong correlation was observed between the Cq values obtained from the reference RT-qPCR assay on the standard laboratory platform and the Cq values derived from PlasmoPod (Fig 3A). The detection probability of PlasmoPod was dependent on the input target molecule number, as indicated by the Cq values obtained from the reference RT-qPCR assay (Fig 3B). At an ultra-high Cq value of 39.9, the estimated detection rate was 64.1% (95% CI: 42.6–85.6%), suggesting a high recall rate even at very low target molecule concentrations.

This dataset encompassed individuals of all age groups, ranging from 1 to 75 years old, which is of particular interest since older asymptomatic individuals are expected to exhibit higher natural immunity and consequently lower parasite densities. To conduct a more detailed analysis, we stratified the cohort into children (up to 15 years old) and adults (Fig 3C). Children exhibited a higher detection rate (88.4%) compared to adults (76.3%), likely due to the typically higher parasite densities observed in children. The Cq values obtained with PlasmoPod were lower in the children's group, although this difference was not statistically significant (Fig 3D).

## Parasitological and clinical characteristics of clinical malaria cohort used for PlasmoPod test evaluation

Next, the performance of the PlasmoPod was evaluated by analyzing blood samples collected from febrile patients admitted to the Paediatric Hospital and University Complex of Bangui. DBS from a total of 47 children were included and an overview of the parasitological and demographic characteristics of these children are shown in **Table 1**. The age of the children ranged from 2 months to 15 years with 48.9% (23/47) being female. DBS collected from these children were screened for *Plasmodium* spp. NAs with a high sensitivity diagnostic RT-qPCR assay based on the parasites' 18S ribosomal DNA/RNA (Pspp18S assay) using the Bio-Rad CFX96 qPCR device. Parasite density in *P. falciparum* positive children was estimated by the PfRNR2 qPCR assay and children were stratified accordingly into moderate (<5000 parasites/

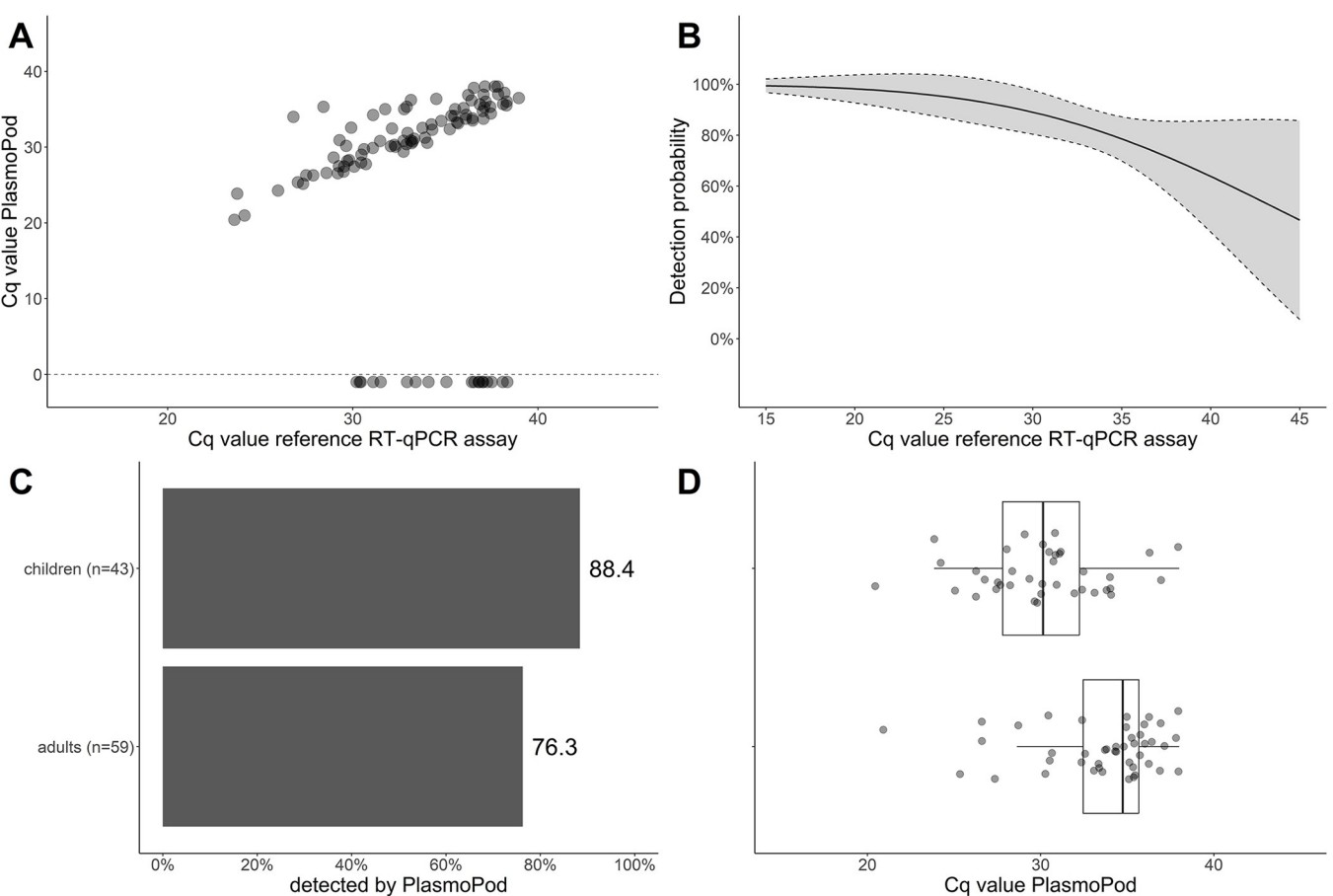

**Fig 3. Analysis of asymptomatic malaria cohort.** (A) Correlation of of Cq valuesvales obtained from reference RT-qPCR run on the Biorad CFX96 instrumentintrument and PlasmoPod run on diaxxoPCR. Samples negative for PlasmoPod were assigned a Cq value of -1. (B) Detection probability for PlasmoPod modelled based on reference Cq values. The grey area represents the 95% confidence interval. (C) Detection rate of PlasmoPod stratified by age group. (D) Cq values stratified by age group.

μL) and high (≥5000 parasites/μL) parasite density groups. Out of 47 children, 16 were negative for *Plasmodium* spp. and *P. falciparum* by RT-qPCR screening, 16 had moderate *P. falciparum* parasite densities and 15 had higher *P. falciparum* parasite density infection. The children assigned to the higher parasite density group were younger compared to the children with moderate parasite densities or children without detectable parasites.

A strong correlation between parasite densities derived from TBS microscopy and qPCR was observed (**Fig 4A**). Interestingly, five out of the seven Pspp18S qPCR-positive samples

**Table 1. Parasitological and demographic characteristics of the study population selected for evaluation of PlasmoPod from Central African Republic.**

| Malaria stratification | Number of children | Age | Sex (% female) | Parasite density (parasites/μL) |
|---|---|---|---|---|
| | | **Median and Range** | | |
| Negative for malaria | 16 | 5.5 years | 50.0% | 0 |
| | | 6 month– 15 years | | |
| Moderate parasite density | 16 | 3 years | 46.7% | Median: 1158 |
| <5000 per μL | | 10 month– 14 years | | IQR: 447–2858 |
| High parasite density | 15 | 13 months | 53.3% | Median: 25'800 |
| >5000 per μL | | 2 month—8 years | | IQR: 12'171–47'960 |

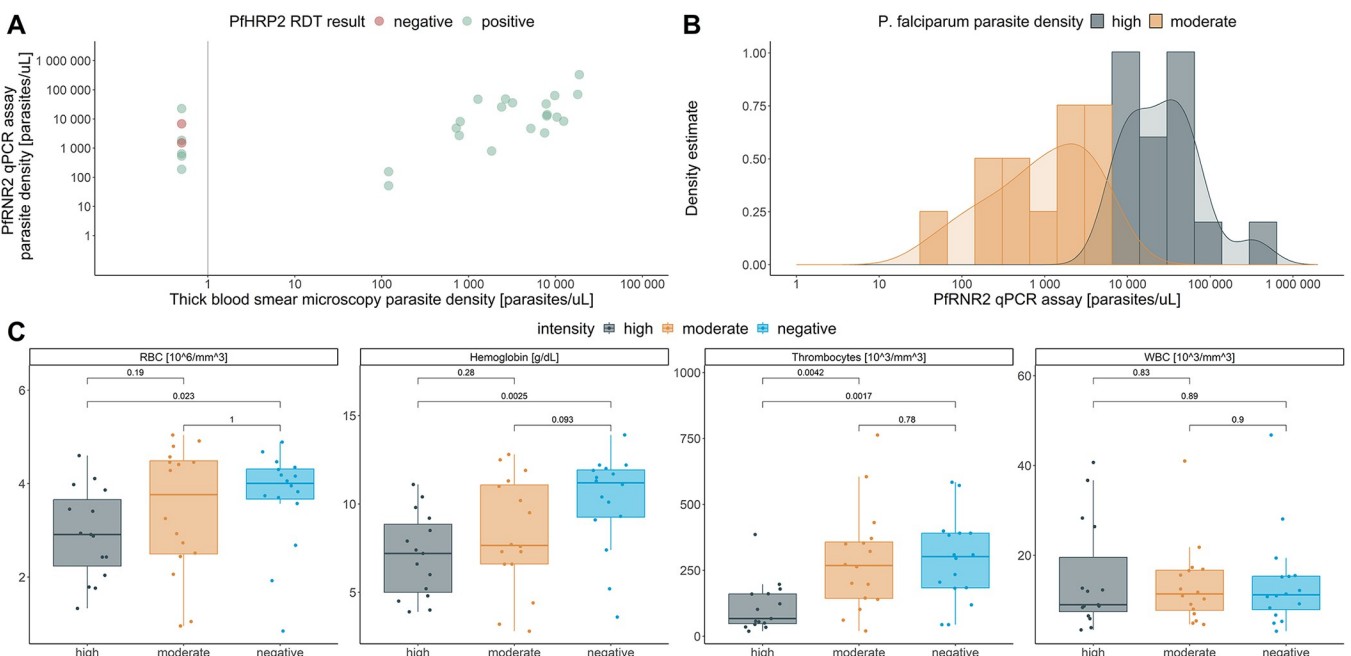

**Fig 4. Clinical and parasitological characterization of study population.** (A) Correlation of parasite density assessed by the PfRNR2 assay and thick blood smear microscopy. (B) Modelled distribution of parasite densities measured by PfRNR2 assay and highlighted for high- and moderate-density infections. (C) Hematological parameters, including red blood cell counts (RBC), Hemoglobin levels (Hb), Thrombocyte counts (PLT) and white blood cell counts (WBC) compared between malaria negative children and children with high- and moderate-density malaria infections.

which were negative by TBS microscopy, were positive by PfHRP2-based RDT, indicating a low diagnostic performance of the TBS microscopy in this setting. The modelled parasite density for the moderate- and high-density groups are visualized in **Fig 4B**. A large dynamic range, covering parasite densities from 52 to 332'983 parasites/µL, is included in this diagnostic test evaluation. Next, we wanted to compare the four hematological parameters, including counts on erythrocytes (RBC), leukocytes (WBC), thrombocytes (PLT) as well as the hemoglobin concentration (Hb) that had been collected from these children during blood collection stratified by malaria infection status (**Fig 4C**). RBC, PLT and Hb were significantly lower among the high parasite density infection group compared to the malaria negative children.

The parasitological characteristics of the malaria positive children was further analyzed by additional RT-qPCR assays in which the *Plasmodium* spp. species as well as the *pfhrp2* and *pfhrp3* deletion status was investigated. The highly sensitive Pspp18S RT-qPCR assay was run as a multiplex assay combined with the internal control of the screening assay, the human *rnasep* gene (**S1 Fig**). Human-derived NAs were found in all 47 samples with an average Cq value of 27.8 and a standard deviation of 1.5, indicating that the nucleic acid extraction procedure worked efficiently and consistently. All 31 malaria positive children were tested positive for *P. falciparum* and no other *Plasmodium* spp. species were found (**S1 Fig**). The *P. falciparum* parasites identified in this cohort were analyzed for the presence of *pfhrp2* and/or *pfhrp3* gene deletions and not a single case of a *P. falciparum* strain with *pfhrp2*/*pfhrp3* gene deletion was found (**S1 Fig**).

## PlasmoPod, coupled with a quick NA extraction procedure from DBS, enables rapid malaria diagnosis

A clinical evaluation dataset, including the well characterized malaria negative and positive samples described above, was used to compare the diagnostic performance of PlasmoPod with

malaria diagnosis based on PfHRP2/panLDH RDTs and TBS microscopy. During the clinical evaluation stage, PlasmoPod was run with NAs extracted by the rapid Chelex-based procedure on the diaxxoPCR instrument. As the gold standard for qualitative comparisons, we used the outcome of the highly sensitive Pspp18S RT-qPCR assay based on amplification of NAs extracted with the NEM protocol and run on the Bio-Rad CXF96 qPCR instrument. All quantitative analysis was conducted based on the parasite densities obtained by the highly accurate PfRNR2 qPCR assay. In comparison to the gold standard the sensitivities and the specificities were calculated summarized in **Table 2**. PlasmoPod showed an overall sensitivity of 93.6%. With 83.9%, the PfHRP2-based RDT achieved the second highest sensitivity, while TBS microscopy and panLDH-RDT outcomes resulted in overall sensitivities below 70%. As expected, all diagnostic methods achieved higher sensitivity in children with high parasite densities compared to children with moderate parasite densities. Among children with moderate parasite densities, PlasmoPod missed 2/16 children resulting in a sensitivity of 87.5% for this group. Interestingly, the two false-negative children had parasite densities below the LOD of the PfRNR2 qPCR and were only detected by the highly sensitive RT-qPCR assay based on amplification of 18S ribosomal total NAs. Sensitivities among the moderate parasite density ranged from 75.0% to 43.8% for the other three diagnostic methods. Only TBS microscopy and PlasmoPod, performed with a 100% specificity. The panLDH-RDT and PfHRP2-RDT tests were wrongly positive in 1, and 2 out of 16 *Plasmodium* spp. negative children, respectively.

A strong correlation of Cq values derived from the PlasmoPod measurements obtained from the di-axxoPCR device and the Cq values obtained from two reference qPCR assays run on the Bio-Rad CFX96 instrument was observed (Fig 5A). The correlation of PlasmoPod with the Pspp18S RT-qPCR wasis stronger than the correlation with the DNA-based PfRNR2 qPCR assay. There wasis an overall high correlation between two independent qPCR assays, run with DBS extracted NAs following standard procedures on a standard qPCR instrument like Bio-Rad CFX96 qPCR instrument, and our novel approach based on rapid extraction procedure from DBS in combination with ready-to-use PlasmoPod cartridgesPlasmoPods and rapid PCR cycling. Additionally, a significant correlation between PlasmoPod Cq values and parasite densities measured by thick blood smear microscopy was observed (Fig 5B). In summarysummay, the data presented is a strong indication that PlasmoPod allows for quantitative measurements of malaria parasites in DBS collected under field conditions.

## Discussion

Accurate and reliable diagnostic tests are the fundamental backbone of healthcare systems. Yet, 47% of the global population has little to no access to diagnostics [30]. The global technical

**Table 2. Diagnostic performance of RDT (PfHRP2/panLDH), TBS microscopy and PlasmoPod compared to the gold standard RT-qPCR assay run on the BioRad CFX96 qPCR instrument.**

| | Sensitivity (95% CI) | | | Specificity (95% CI) |
|---|---|---|---|---|
| | All positive children | Moderate-density infections | High-density infections | - |
| | - | (<5000 parasites/μL) | (>5000 parasites/μL) | - |
| | n = 31 | n = 16 | n = 15 | n = 16 |
| TBS microscopy | 64.55% (20/31) (45.4% - 80.8%) | 43.8% (7/16) (21.3% - 73.4%) | 86.7% (13/15) (59.5% - 98.3%) | 100.0% (16/16) (79.4% - 100.0%) |
| PfHRP2-RDT | 83.9% (26/31) (66.3% - 94.6%) | 75.0% (12/16) (47.6% - 92.7%) | 93.3% (14/15) (68.1% - 99.8%) | 87.5% (14/16) (61.7% - 98.5%) |
| panLDH-RDT | 67.7% (21/31) (48.6% - 83.3%) | 62.5% (10/16) (32.3% - 83.7%) | 80.0% (12/15) (51.9% - 95.7) | 93.8% (15/16) (69.8% - 99.8%) |
| PlasmoPod | 93.6% (29/31) (78.6% - 99.2%) | 87.5% (14/16) (61.7% - 98.5%) | 100.0% (15/15) (78.2% - 100.0%) | 100.0% (16/16) (79.4% - 100.0%) |

CI = Confidence interval

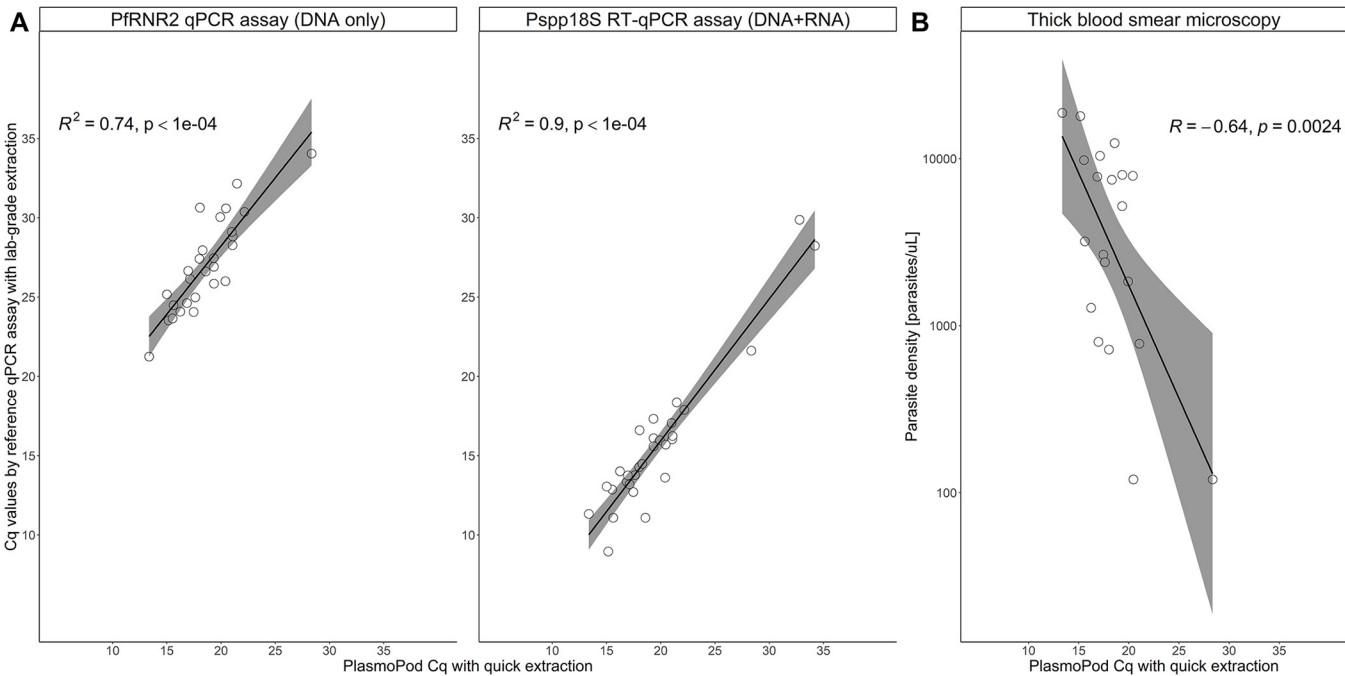

**Fig 5. Quantification of *P. falciparum* parasites using PlasmoPod.** (A) Correlation of Cq values derived from PlasmoPod and reference (RT)-qPCR assays based on *Plasmodium* spp. 18S ribosomal DNA and RNA (left panel) and the *P. falciparum* ribonucleotide reductase R2_e2 assays (right panel). (B) Correlation between PlasmoPod and thick blood smear microscopy for quantification of *P. falciparum* parasites. The grey color represents the 95% confidence interval.

strategy for malaria 2016–2030 sets the target of reducing global malaria incidence and mortality rates by at least 90% by 2030 [31]. One of the major pillars of the strategy ensuring access to malaria prevention, treatment and diagnosis. Expansion of diagnostic testing is required to provide timely and accurate surveillance data. This data is crucial for tracking the successes or drawbacks of malaria control and elimination efforts. Furthermore, in low transmission settings aiming for malaria elimination, the large-scale deployment of highly sensitive and specific diagnostic techniques is required to ensure the accurate diagnosis of low density asymptomatic parasite carriers [32]. Due to limited access to sensitive molecular tests for malaria surveillance and despite their low diagnostic performance, PfHRP2-based RDTs are still the most widely used diagnostic tests for malaria surveillance in endemic regions [2]. Molecular diagnostic techniques, in particular PCR-based tests, are much more accurate tools for surveillance. Incorporating NAATs in reactive case detection (RCD) [33] or monitoring activities related to mass drug administration (MDA) [34] programs provides clear advantages over antigen-based RDTs due to their higher sensitivity. However, these diagnostic tests are rarely used in malaria control programs since the infrastructure is often limited to centralized testing facilities and the costs of equipment and consumables are relatively high [35]. Traditional approaches to molecular malaria surveillance are centralized, where samples are collected and analysed at reference laboratories. This can be limit the process by logistical and financial constraints, leading to under-detection and under-reporting of malaria cases. Decentralized testing, on the other hand, involves the use of portable diagnostic devices and decentralized laboratories. These can be deployed at the point-of-care or in community settings. This approach enables more rapid and widespread testing, leading to more timely and accurate disease surveillance. In addition, decentralized testing can reduce the burden on central laboratories and improve access to testing in underserved or remote areas.

As a first step towards our goal of developing novel tools for improved and decentralized malaria surveillance, we designed, developed and extensively evaluated a rapid NAAT-based diagnostic test for *Plasmodium* spp. parasites using the portable and low-cost diaxxoPCR platform. Starting from DBS, in less than 30 minutes a diagnostic test for malaria is conducted with an analytical performance similar to sensitive, state-of-the-art laboratory-based RT-qPCR assays. Our novel approach is based on PlasmoPod cartridges which requires little hands-on time, no cold chain and almost no technical expertise as they are preloaded with all RT-qPCR reagents. In this initial study, we introduced a proof-of-concept for a cartridge-based molecular malaria test and presented validation data for our PlasmoPod platform. However, we acknowledge that further validation of this platform is necessary through direct testing conducted in endemic countries. A major limitation of our study is that the validation was conducted in a controlled laboratory setting, which does not necessarily represent the real-world conditions found in malaria-endemic testing sites. Additionally, we have identified several areas that can be improved. The current process of rapid extraction and loading of the PlasmoPods still relies on manual pre-processing of the samples. The pipetting steps involved present a challenge for widespread implementation of our approach, as it requires a certain level of expertise. Therefore, the next generation of the diaxxoPCR platform should incorporate automated nucleic acid extraction and loading of the PlasmoPods to reduce hands-on time and manual sample handling. By overcoming these challenges, PlasmoPods in combination with rapid PCR cycler could be well-suited for deployment to satellite public health laboratories enabling decentralization of molecular malaria surveillance activities or for remote health care settings which do not have a fully equipped laboratory infrastructure.

In this initial study, we used a three-step approach to evaluate PlasmoPod for malaria diagnosis. We started with an analytical performance evaluation using well characterized samples and laboratory strains. Two different samples sets originating from two Central Africa countries, the Central African Republic and Equatorial Guinea, were used to further test the performance of PlasmoPod. Samples from malaria asymptomatic individuals from Bioko Island, Equatorial Guinea and symptomatic children from Bangui, Central African Republic, were analysed with PlasmoPod.

Our analytical performance evaluation suggests that a LOD as low as 0.02 parasites/µL can be achieved if NA are extracted from whole blood and no cross-reactivity with other pathogens common in malaria endemic regions was observed. This high sensitivity is achieved by using the highly abundant *Plasmodium* spp. 18S ribosomal total NA (RNA and DNA) as a biomarker for malaria infection. The parasite's 18S ribosomal NAs are present as a multicopy gene in the DNA as well as transcribed as RNA molecules.

PlasmoPod exhibited an 81.4% detection rate when analysing samples from 102 asymptomatic malaria-positive individuals on Bioko Island, emphasizing its efficacy as a sensitive molecular tool for malaria surveillance, particularly among asymptomatic cases.

The evaluation of the PlasmoPod and diaxxoPCR for malaria diagnosis among symptomatic children was performed with DBS collected from children attending the emergency department at the Paediatric Hospital and University Complex of Bangui. The results revealed that the PlasmoPod has a sensitivity of 100% if tested with children having high parasite densities and 87.5% if tested with children having a moderate-density infection. Since the gold-standard test used purified nucleic acids extracted from an entire DBS corresponding to approximately 30 µL of blood, while the PlasmoPod approach utilized only a 3 mm DBS punch containing 1–2 µL of blood [36], we conclude that PlasmoPod is likely to achieve a performance similar to a laboratory-based diagnostic platform for diagnosing symptomatic patients. Using a highly abundant biomarker like 18S ribosomal RNA and DNA for the RT-qPCR, the lower sample input and lack of highly pure NA due to the rapid extraction process,

can be compensated resulting in a robust approach suitable for field applications. Furthermore, it was shown that, diagnostic performance of the TBS microscopy and the PfHRP2/panLDH-RDTs lack sensitivity. Even among the 15 symptomatic children carrying parasites densities above 5'000 parasites/μL, two were missed by TBS microscopy, one by PfHRP2-RDT and three by the panLDH-RDT. In addition to lack of sensitivity, also the specificity of the RDTs was reduced. Two out of 16 *Plasmodium* spp. negative children were wrongly tested positive for *P. falciparum* by PfHRP2-based antigen RDT. False-positive RDTs are common in endemic regions and are likely caused by persisting PfHRP2 antigen circulation post anti-malarial treatment [37, 38]. However, a limitation of our study was that we used an RDT that had not undergone WHO prequalification.

In summary, using samples from two different independent cohorts, including asymptomatic individuals and symptomatic patients, PlasmoPod achieved sensitivities above 80% compared to a highly sensitive RT-qPCR assay. While the initial evaluation of PlasmoPod using samples from 149 individuals shows promising results, further validation studies are necessary to fully assess its performance and reliability. During this initial testing phase, PlasmoPods underwent successful evaluation using NAs extracted through various methods. These methods encompassed commercial column-based extraction kits, the ENAR protocol from archived RDTs, and chelex-based extraction from DBS samples. While the method of NA extraction influences overall sensitivity, it's noteworthy how PlasmoPod demonstrates efficacy across varying levels of NA quantity and purity.

Over the past decade, various NAATs for malaria diagnosis have been published, utilizing different technologies such as qPCR [18, 21, 39–42] and isothermal amplification [43–48]. Each of these assays has its own advantages and disadvantages concerning analytical performance, throughput, simplicity, storability, and laboratory setup requirements. In this study, we introduced the PlasmoPod as a proof-of-concept for a cartridge-based NAAT, aiming to simplify and standardize molecular malaria diagnosis and surveillance. Currently, the PlasmoPod is designed to detect conserved NA sequences present in all human pathogenic *Plasmodium* spp. species. However, future development of this platform should focus on the design and validation of cartridges specifically designed for identifying individual *Plasmodium* spp. species. Given that the diaxxoPCR device is a universal qPCR platform, adapting published multiplex species-specific qPCR assays [23, 49, 50] to the PlasmoPod could be a straightforward process. Additionally, separate cartridges could be developed for the molecular characterization of *P. falciparum* isolates. Incorporating assays that enable the detection of *pfhrp2* gene deletions [24, 51, 52] or molecular markers of anti-malarial drug resistance [53, 54] could be particularly valuable in supporting decentralized malaria surveillance efforts. By expanding the capabilities of the PlasmoPod platform to include species identification and molecular characterization of drug resistance, we can enhance its utility and contribute to more effective malaria control and surveillance programs.

## Conclusions

In conclusion, we have established a 30-minute sample-to-result RT-qPCR procedure that delivers results with similar diagnostic performance as state-of-the-art RT-qPCR assays for malaria diagnosis. In most malaria endemic regions, molecular malaria diagnostics are only available in centralized laboratories and inaccessible at peripheral health facilities where they are needed most. We believe that the PlasmoPod rapid NAAT can bridge this gap and will enable widespread accessibility of high-quality, sensitive and easy to handle molecular malaria testing at the individual as well as the population level allowing decentralization of testing and surveillance activities.

## Supporting information

**S1 Fig. Molecular analysis of *Plasmodium* spp. parasites identified in the CAR dataset.**
Three different molecular assays were used to (A) screen for *Plasmodium* spp. parasites, (B)
identify *Plasmodium* spp. species and (C) detect *pfhrp2/3* gene deletion. Each child is represented in a column stratified according to malaria infection status. Green colors represent negative measurements for the respective qPCR assay, while grey colors were chosen for tests
which were not conducted. All tests were run on the Bio-Rad CXF96 qPCR instrument.
(TIF)

**S1 Data. Supplementary dataset.** All data that support the findings of this study are available
as a supplementary document uploaded to the journal's website.
(XLSX)

## Acknowledgments

The authors would like to thank the administrative and laboratory staff of the Paediatric Hospital and University Complex of Bangui (CHUPB) for their support and fruitful cooperation.
The authors would also like to thank all patients and their legal guardians who participated in
this study. We would also like to thank A&B Professional company located in Lucca, Italy
which donated the malaria RDTs used in this study. The authors would also like to acknowledge the contribution of the technical and scientific personnel of Medical Care Development
Global Health conducting the yearly malaria indicator surveys on Bioko Island.

## Author Contributions

**Conceptualization:** Philippe Bechtold, Michele Gregorini, Tobias Schindler, Ulrich Vickos.

**Data curation:** Olivier Tresor Donfack, Tobias Schindler, Ulrich Vickos.

**Formal analysis:** Philippe Bechtold, Philipp Wagner, Tobias Schindler.

**Funding acquisition:** Wendelin J. Stark, Wonder P. Phiri, Guillermo A. García.

**Investigation:** Jean Chrysostome Gody, Marilou Sonia Pagonendji, Olivier Tresor Donfack,
Ulrich Vickos.

**Methodology:** Philippe Bechtold, Philipp Wagner, Salome Hosch, Edwige Régina Kodia-Lenguetama, Marilou Sonia Pagonendji, Christian Nsanzanbana, Ulrich Vickos.

**Project administration:** Tobias Schindler.

**Resources:** Wendelin J. Stark, Edwige Régina Kodia-Lenguetama, Olivier Tresor Donfack,
Wonder P. Phiri, Guillermo A. García, Christian Nsanzanbana, Claudia A. Daubenberger.

**Supervision:** Claudia A. Daubenberger, Tobias Schindler, Ulrich Vickos.

**Validation:** Philippe Bechtold, Michele Gregorini, Tobias Schindler.

**Visualization:** Philippe Bechtold, Philipp Wagner, Salome Hosch.

**Writing – original draft:** Philippe Bechtold, Claudia A. Daubenberger, Tobias Schindler.

**Writing – review & editing:** Philippe Bechtold, Claudia A. Daubenberger, Tobias Schindler.

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
