## [Decision Letter · Decision Letter 0]

10 Feb 2023

PGPH-D-22-02106

Development and evaluation of PlasmoPod: A cartridge-based nucleic acid amplification test for rapid and decentralized malaria diagnosis and surveillance

Dear Dr. Schindler,

Thank you for submitting your manuscript to PLOS Global Public Health. After careful consideration, we feel that it has merit but does not fully meet PLOS Global Public Health’s publication criteria as it currently stands. Therefore, we invite you to submit a revised version of the manuscript that addresses the points raised during the review process.

We look forward to receiving your revised manuscript.

Kind regards,

Sarah Auburn

Academic Editor

Journal Requirements:

1. Please send a completed 'Competing Interests' statement, including any COIs declared by your co-authors. If you have no competing interests to declare, please state "The authors have declared that no competing interests exist". Otherwise please declare all competing interests beginning with the statement "I have read the journal's policy and the authors of this manuscript have the following competing interests:"

a. State what role the funders took in the study. If the funders had no role in your study, please state: “The funders had no role in study design, data collection and analysis, decision to publish, or preparation of the manuscript.”

b. If any authors received a salary from any of your funders, please state which authors and which funders.

3. Please provide separate figure files in .tif or .eps format only and remove any figures embedded in your manuscript file. Please also ensure that all files are under our size limit of 10MB.

4. We do not publish any copyright or trademark symbols that usually accompany proprietary names, eg  ©, ®, ™  (e.g. next to drug or reagent names). Please remove all instances of trademark/copyright symbols throughout the text, including ® on page 9.

5. In the online submission form, you indicated that "The data that support the findings of this study are available from the authors upon reasonable request". All PLOS journals now require all data underlying the findings described in their manuscript to be freely available to other researchers, either 1. In a public repository, 2. Within the manuscript itself, or 3. Uploaded as supplementary information.

Additional Editor Comments (if provided):

The manuscript was reviewed by three independent experts. All three reviewers identified the study results as potentially important for the malaria diagnostics field, but only with more detailed information on the methods, further description of the scope to introduce PlasmoPod into remote public health care settings, and a transparent discussion on the limitations of the study design and product. Please pay particular attention to the following requests and concerns raised by the reviewers:

1. Request for supplementary data for full transparency

2. Request for more information on the product including cost, throughput, and potential resourcing/supply challenges

3. Concerns about lack of testing in a malaria-endemic setting and robustness of associated claims about this potential

4. Request for clarity on the study design, including the small sample size

5. Concerns about the lack of low density infections

6. Concerns about the Hb level diagnostic approach

7. Request for more detail on the scope for species-specific diagnosis

Reviewers' comments:

Reviewer's Responses to Questions

**Comments to the Author**

1. Does this manuscript meet PLOS Global Public Health’s publication criteria? Is the manuscript technically sound, and do the data support the conclusions? The manuscript must describe methodologically and ethically rigorous research with conclusions that are appropriately drawn based on the data presented.

Reviewer #1: Yes

Reviewer #2: Yes

Reviewer #3: Yes

2. Has the statistical analysis been performed appropriately and rigorously?

Reviewer #1: No

Reviewer #2: Yes

Reviewer #3: Yes

3. Have the authors made all data underlying the findings in their manuscript fully available (please refer to the Data Availability Statement at the start of the manuscript PDF file)?

Reviewer #1: No

Reviewer #2: Yes

Reviewer #3: Yes

4. Is the manuscript presented in an intelligible fashion and written in standard English?

Reviewer #1: Yes

Reviewer #2: Yes

Reviewer #3: Yes

5. Review Comments to the Author

Reviewer #1: Bechtold and colleagues tested a novel, portable qPCR device for P. falciparum diagnosis. They also evaluate of a rapid DNA/RNA extraction protocol, which is equally crucial for low-resource settings. They evaluated the method on cultured parasites, and on a small set of clinical samples. While the novel method has the potential to improve malaria surveillance, several points need to be addressed before publication.

Main comments:

1) The information on PlasmoPod/diaxxoPCR is quite sparse. There is a miniature figure of the instrument as part of figure 1. Please add a larger figure, and also add a figure of the chip.

What is meant by “next generation qPCR instrument” (line 65)? How many samples can be run in parallel? Can samples and a standard curve be run on the same chip, or is this not needed? Can the same assay be run on other portable qPCR instruments, or is it limited to the diaxxoPCR (e.g. the one tested here: https://journals.plos.org/globalpublichealth/article?id=10.1371/journal.pgph.0000454)?

What is the cost of the instrument, and what is the per sample cost?

2) The main limitation is the small and biased dataset of only 47 field samples used to evaluate the new PCR. The lowest parasite is 54 parasites/uL, i.e. much higher than what would be observed in a real-world setting. How were these samples selected? Why were no samples with lower density included?

3) Following the point above, the 95% detection probability based on a probit analysis of the field samples is included. Such an analysis requires a large number of samples around the LOD of each method. Given the very limited number of samples available, the results are unreliable and completely implausible. For example, the 95% detection probability for HRP2-based is calculated as 11,000 parasites/uL. This is approx. 100-fold higher than other studies that tested HRP2-based RDTs (as correctly stated in line 56). While differences are likely due to different RDTs used, study population characteristics (i.e. different mean parasite densities), and technical variation of density calculation based on qPCR, such large differences are implausible.

Unless the authors are able to run many more samples (in particular low density samples), this analysis cannot be conducted.

4) The authors divide their samples in high density (>5000 parasites/uL) and low density (<5000 parasites/uL). This seems like an odd choice. RDTs now reach limits of detection of <100 parasites/uL. The samples included in the low density groups are in no way low density with respect of diagnosis. Almost all of them will easily be detected by RDT. These groups should be combined, and references to low-density samples removed, as there are almost none.

5) Lines 190-205: Was the cultured blood spotted on DBS, and a single 3mm punch used for extraction? If so, it is difficult to understand how a positive result can be obtained at 0.02 parasites/uL. One DBS punch is expected to correspond to 1-2 uL of blood (line 353). Assuming a single parasite was present in 2 uL, the theoretical lowest LOD will be 0.5 parasites/uL.

If for this test DNA+RNA was extracted from whole blood, please add this info. It might also be worth do add sentence or two to the discussion explaining the importance of this difference (lines 401-402)

The situation would of course be different if the target RNA was secreted to the blood. If this is the case, please describe.

6) The authors develop a threshold based on hemoglobin levels as diagnostic marker for malaria and compare the new PCR against it. While the relationship between low Hb levels and infection have long been established, it is well known that the correlation is at best moderate. It is very odd to compare a new test to an approach that was developed ad hoc based on a very small dataset, is not used by control programs, and not recommended by the WHO. This analysis should be removed.

7) For each sample (cultured and clinical), all data including Cq value by conventional and PlasmoPod qPCR, parasite density by microscopy (for clinical samples), and RDT results (for clinical samples), should be included as a supplementary file. The clinical data can be presented complete de-identified. There is no reason this data is only available from the authors upon reasonable request. This is of particular importance as several co-authors are employed by the company producing the PlasmoPod and thus have a commercial interest.

8) I commend the authors for keeping their discussion short and to the point!

Minor comments:

Lines 59-62: Re-phrase this sentence, as it implies that well-equipped centralized laboratories are only available in in resource-constraint settings.

Line 68: What is meant by ‘Control strategy’? Should that read amplification strategy?

Lines 74-76: Please provide a reference

While the technology has the potential for decentralized diagnosis, for the current study all samples were tested at the Swiss Tropical and Public Health Institute. Thus, the term ‘decentralized’ should not be included in the title.

Line 152: Typo: diaxx-oPCR

Figure 4: For added clarify, please add what green and grey means to the little plot at the right showing the color coding of the Cq values.

Also, the figure might be easier to read if the height of the boxes was reduced.

Reviewer #2: Bechtold et al., describe PlasmoPod, a cartridge-based nucleic acid (NA) amplification test ran on diaxxoPCR platform for the detection of Plasmodium spp. NA is rapidly extracted from DBS, and the PCR is completed in less than 30 minutes. The performance of this device is compared to commercially and well-established molecular detection platforms with established extraction and NA amplification and detection methods. The diaxxoPCR platform has recently been described in the development of other cartridge-based NA amplification tests for other infectious diseases including the detection of SARS-CoV-2. The authors use previously published Plasmodium primer sets to create PlasmoPod. In this study, the authors describe the analytical performance of the PlasmoPod using Plasmodium NA obtained from cultures, the WHO International Standard for P. falciparum DNA for NAAT-based assays, and field collected samples. The designed, execution and the analysis of the study is well thought through, and some of the analyses performed are important in assessing the performance of the PlasmoPod device. I have some comments for the authors.

MAJOR Comments

1. The PlasmoPod and diaxxoPCR were used to evaluate a previously developed single target malaria PCR. The test was performed in a highly controlled environment, probably by the platform and/or test developers. The authors cannot claim that this test/platform is suited for deployment to satellite public health laboratories or remote health care settings since they had an opportunity to test the performance of the platform in the field settings in CAR, but they choose to bring the specimen back to Switzerland. Until they deploy the test/platform in the field settings where they think the platform is suited, then right now all they can do is speculate. The test/platform can analyze NA from arguably a straight forward, and simple protocol. However, in clinical labs and other low or limited resource (including manpower) setting, sample extraction process is a limiting factor when compared to a platform such as the GeneXpert, a cartridge-based platform which requires less than 2 minutes hand-on sample loading and includes sample extraction. It is good to remember that in a clinical lab setting, many different tests are run at the same time by many technicians, and that’s why what might seem straight forward when developing a test in a highly controlled environment might produce different outcome in high throughput or resource limited settings.

2. Using hemoglobin value as a standalone malaria diagnostic parameter has no clinical relevance and is absolutely misleading since hematological parameters can be impacted by many factors including other infectious diseases, clinical or genetical conditions. However, hemoglobin and platelet can be important parameters in understanding severity of malaria disease, and along with other specific diagnostic methodologies for the detection of malaria parasite such as those described here (molecular, antigen detection), hematological parameters (hemoglobin and platelet) can provide an additional layer of information or algorithm that can be used to increase diagnostic test specificity. The authors must revisiting/rewriting and reanalyzing hemoglobin as a diagnostic parameter since it cannot be used for malaria diagnosis as a standalone diagnostic tool, but as a tool that can complement other methods.

Other comments

1. There are other malaria tests available as analyte specific reagent (ASR) and can be available for most PCR platforms already in the market such as this one https://www.ncbi.nlm.nih.gov/pmc/articles/PMC4026594/. The authors should discuss the performance of their test/platform and compare the performance to other platforms in the market.

2. The PlasmoPod has only a single target, Plasmodium at genus level. Malaria epidemiology is shifting due to elimination efforts and climate chance, and places such as Ethiopia and Somalia have two different malaria species circulate that require different treatment options. The authors must address this as one of the limitations of this test and how do they envision overcoming this without making testing increasingly complicated and expensive.

3. Not all diagnostics tests are perfect, but there must be a balance such as the ease of use, reagent and platform cost, the need for infrastructure etc. PlasmoPod sensitivity is same as that of PfHRP2-based RDT. The authors have not addressed reagent and platform cost. This is important, one of the main reasons why molecular tests are still not widely deployed for malaria testing. For marginal increase in sensitivity, the authors should present strong argument why this test should be considered including low cost for the reagents and platform (if this is true; must compare cost to current platforms in the market of the same class/caliber), and potential for low maintenance etc. The authors must remember GeneXpert has penetrated the low resource settings mostly because of the TB program, and the infrastructure including the supply chain. How do the authors see such a platform penetrating the market including having readily available supply chain?

4. The authors have not addressed any study limitations including the study design and the results obtained.

5. The authors describe three different PCR methods, with limited or scanty details, mostly referencing previously published methods. They also have used different specimen types (e.g., DBS extracted using different protocols etc.). It is a little hard to follow as they describe these tests back and forth but what is even more frustrating for example, starting lines 346-361, they attempt to describe how PlasmoPod performs equally as good as the other PCR methods, using details in the method that we either referenced or not chronologically well presented.

Reviewer #3: This manuscript reports the evaluation of a newly developed molecular diagnostic for malaria infection. It includes an assessment of the LOD by using a serial dilution of a cultured P. falciparum sample; tests for cross-reactivity with other commonly circulating pathogens; and finally a performance evaluation with samples from febrile patients seeking care at a hospital in Central African Republic. Diagnostic performance is compared to two standard and high-sensitivity molecular PCR-based assays. The diagnostic performance of the new PlasmoPod assay looks very impressive and encouraging. Some discussion of the next steps and further work needed to confirm these results and further evaluate the assay would be important to include, e.g. in the field across diverse patient groups and malaria species. The assay is currently not species-specific. Any scope to address this should be described in the paper.

More description of diaxxoPCR device itself and the PlasmoPod cartridges would be useful to give an idea of how this tech could feasibly be used: e.g. cartridge storage restrictions, fragility (how feasible to transport in back of a truck?), size and weight, indicative cost range, throughput (samples per hour).

The “Hb-8.8” method is not standard practice. I would strongly recommend excluding from this evaluation; it does not add anything.

Introduction:

- Line 52: 419m RDTs sold in 2020 was the estimate of PQ-approved assays only, could add that detail

- Line 54: Reference 7 dates to 2006 and RDTs have improved in quality since then. A more recent evaluation of RDT performance to check: https://apps.who.int/iris/bitstream/handle/10665/276193/9789241514958-eng.pdf?ua=1

- Line 71: car battery: can other types of batteries also be used? Are these included as part of the kit? Can the assay also be connected to the mains power supply?

- Mention how conserved the 18S ribosomal DNA and RNA are: would these be expressed by zoonotic species as well as all 5 human Plasmodia?

Methods:

- Suggest moving Figure 1 into Methods section.

- Line 88: “admitted to the emergency department”. How were patients selected for inclusion?

- Contradiction in the RDT brand used in the study: CareStart (line 90) or A&B Professional (line 463). Which was used? Note that CareStart was issued a WHO Notice of Concern in 2021; A&B Professional has not yet received WHO Pre-qualification for their assays. Any future evaluations should prioritise WHO PQ-approved RDTs if possible. Could refer to this limitation in the Discussion.

- Line 91: given the results with microscopy, worth stating the experience of the microscopists reading the slides. Were slides double-read?

- Need to add target sample size and justification for this.

- Line 102: write in full “NEM”

- Line 124: the cut-off for high/low parasitaemia of 5000p/ul seems arbitrary and convenient given the samples collected. Justify this threshold, especially if the assay’s main strength is detection of very low density infections. A cut-off of 2000p/ul would have been consistent with the upper threshold from the WHO RDT evaluation protocols.

- Line 132: “amplification and detection of Plasmodium spp”: possible to give more detail on the type of technological approach that’s used?

- Line 149: how is the heating of the DBS in Chelex to 95C done in the field? A water bath would be needed?

Results:

- Fig 3B: could the y-axis scale be absolute numbers of samples?

- Fig 4: mention pfmr2 in the legend; what Cq value was used as the cut-off for the green?

- Line 290: specify P. falciparum in the header here: no other species was evaluated

- Lines 301 and 305: would compare like-with-like: quote figures from the same sample groups

- Table 2: add in absolute numbers

- Fig 5: Interesting figure/analysis. Possible to explain the modelled >50% detection probability of the HRP2 RDT at such low densities? The specificity doesn’t seem to explain completely.

- Lines 353-354: suggest moving this sentence to end of paragraph (though actually would fit into Discussion section better)

- Line 361: claim about cost-effectiveness needs a basis.

- Fig 6: add unit (Cq) to y-axis title; put p-value in simpler format (e.g. <0.001)

Discussion:

- Line 401: “indicates”; the LOD was determined with a small number of replicates, recommend “suggests”?

- Line 409: “lower parasite densities” would instead say “<5000 p/ul”.

- This initial evaluation gives very encouraging results, but it would be important to discuss the limits of the evaluation and the next steps needed to further understand the assay performance, especially for its target applications.

- Also important to discuss the assay limitations: PlasmoPod detects Plasmodium NA, but does not differentiate species. What scope to develop a species-specific assay? Are there similar high-copy number potential markers that could be used? And any scope for an application to screen for drug resistance markers, even if this had a lower sensitivity, parasitaemia in patients would usually be higher than what’s required for detection of asymptomatic infections.

- PlasmoPod is described as “Cost-effective” a few times. Evidence of this in the context of a specific application is needed to support this statement.

- Giving a target cost range for the test would also be very useful; what does “low cost” mean (line 393)

- LAMP is another portable molecular diagnostic: some reference and comparison should be made to this.

6. PLOS authors have the option to publish the peer review history of their article (what does this mean?). If published, this will include your full peer review and any attached files.

**Do you want your identity to be public for this peer review?** For information about this choice, including consent withdrawal, please see our Privacy Policy.

Reviewer #1: **Yes: **Cristian Koepfli

Reviewer #2: **Yes: **Edwin Kamau

Reviewer #3: No

---

## [Decision Letter · Decision Letter 1]

29 Aug 2023

PGPH-D-22-02106R1

Development and evaluation of PlasmoPod: A cartridge-based nucleic acid amplification test for rapid malaria diagnosis and surveillance

Dear Dr. Schindler,

Thank you for submitting your manuscript to PLOS Global Public Health. After careful consideration, we feel that it has merit but does not fully meet PLOS Global Public Health’s publication criteria as it currently stands. Therefore, we invite you to submit a revised version of the manuscript that addresses the points raised during the review process.

As highlighted by Reviewer 1, the manuscript is close to publication-ready but still requires a few clarifications to ensure that other readers can readily follow and interpret the results. Please address the suggestions raised by Reviewer 1 in the Comments to Author section. These are minor suggestions and will hopefully be quick to address.

We look forward to receiving your revised manuscript.

Kind regards,

Sarah Auburn

Academic Editor

Journal Requirements:

Additional Editor Comments (if provided):

Reviewers' comments:

Reviewer's Responses to Questions

**Comments to the Author**

1. If the authors have adequately addressed your comments raised in a previous round of review and you feel that this manuscript is now acceptable for publication, you may indicate that here to bypass the “Comments to the Author” section, enter your conflict of interest statement in the “Confidential to Editor” section, and submit your "Accept" recommendation.

Reviewer #1: (No Response)

Reviewer #2: All comments have been addressed

2. Does this manuscript meet PLOS Global Public Health’s publication criteria? Is the manuscript technically sound, and do the data support the conclusions? The manuscript must describe methodologically and ethically rigorous research with conclusions that are appropriately drawn based on the data presented.

Reviewer #1: Yes

Reviewer #2: Yes

3. Has the statistical analysis been performed appropriately and rigorously?

Reviewer #1: Yes

Reviewer #2: Yes

4. Have the authors made all data underlying the findings in their manuscript fully available (please refer to the Data Availability Statement at the start of the manuscript PDF file)?

Reviewer #1: Yes

Reviewer #2: Yes

5. Is the manuscript presented in an intelligible fashion and written in standard English?

Reviewer #1: Yes

Reviewer #2: Yes

6. Review Comments to the Author

Reviewer #1: The authors have addressed my comments well. My main remaining major comment is that sometimes the structure of the manuscript makes it a bit hard to follow the main findings. Several different types of sample (culture, archived RDTs, DBS), study populations (clinical, asymptomatic), and extraction protocols were studied, it sometimes is hard to grasp what exactly was done. Some suggestions for improvement:

Lines 240-259: The previous paragraph describes the simplified Chelex extraction, but I believe this analysis was done on DNA/RNA extracted using another method. It would help to clarify that this analysis does not include the impact of the extraction protocol used.

Lines 267-268: It could help to replace ‘samples’ with ‘archived RDTs’ in the title to highlight that another sample type than elsewhere was used. Likewise, the title on lines 298-299 could include the term ‘DBS’. It would also be good to repeat that different extraction methods were used for RDTs and DBS, which likely has an impact on sensitivity.

Lines 269-276: It might be good to repeat here that these 102 samples were selected based on a previous PCR screening. Else the reader might be confused on why all of them are positive.

Lines 298-374: Given the clear aim of this study is to evaluate the PlasmoPod, I find it quite confusing that after the data obtained by the RT-qPCR on the BioRad instrument is presented, a very long paragraph describes demographic patterns and data on RBCs, Hb etc, all of which is not directly relevant for the main aim of the paper. The PlasmoPod results are only given in a subsequent paragraph. Shouldn’t the BioRad vs. PlasmoPod comparison be given first, with the secondary results presented later? (RBC, Hb and similar data could even be put in a supplementary file).

Throughout the manuscript, the use of the terms PlasmoPod, diaxxoPCR, and Pspp18S RT-qPCR are used interchangeably which makes it harder to follow the text. For example, lines 357-361:

“During the clinical evaluation stage, PlasmoPod was run with NAs extracted by the rapid Chelex-based procedure on the diaxxoPCR instrument. As the gold standard for qualitative comparisons, we used the outcome of the highly sensitive Pspp18S RT-qPCR assay based on amplification of NAs extracted with the NEM protocol and run on the Bio-Rad CXF96 qPCR instrument.” From this text, it is not immediately clear that the Pspp18S was used on both platforms. The text could be clarified by either stating in the methods that the Pspp18S assay was always used except when a single copy assay was used for quantification, and then not repeat ‘Pspp18S’, or it should always be used.

Minor comments:

Typo on line 63: P. falciparum and pfhrp2 not in italics

Typo on lines 85-86: diaxx-oPCR

The Pspp18S qPCR assay is first mentioned on line 107, but only on lines 131-132 it is stated that “18S ribosomal DNA and RNA molecules were targeted [18,21] and detected by

a highly-sensitive RT-qPCR (herein referred to Pspp18S RT-qPCR assay) [19]”. It would be more appropriate to include this information when the assay is mentioned first.

My comment on the LOD depending on the amount if input material might not have been fully clear. Line 448 states an LOD of 0.02 parasites/uL. This is correct when NAs are extracted from whole blood. Yet, given this study included many different sample types, it’s a bit misleading. When 5 uL blood are put on an RDT and then NAs are extracted, the assay can only be positive if at least one parasite was present. Thus, the mathematically possible lowest LOD for this assay is 0.2 parasites/uL (1 parasite in 5 uL blood). The sentence on line 448 should be rephrased to include that this LOD can be achieved for whole blood, but will be higher for RDTs and DBS.

Line 519: No diagnosis is needed for MDA. This sentence is confusing, I don’t think MDA should be mentioned here.

Cristian Koepfli

Reviewer #2: No further comments, thank you for adequately addressing all the concerns.

7. PLOS authors have the option to publish the peer review history of their article (what does this mean?). If published, this will include your full peer review and any attached files.

**Do you want your identity to be public for this peer review?** For information about this choice, including consent withdrawal, please see our Privacy Policy.

Reviewer #1: **Yes: **Cristian Koepfli

Reviewer #2: **Yes: **Edwin Kamau

---

## [Editor Report · Decision Letter 2]

5 Sep 2023

Development and evaluation of PlasmoPod: A cartridge-based nucleic acid amplification test for rapid malaria diagnosis and surveillance

PGPH-D-22-02106R2

Dear Schindler,

We are pleased to inform you that your manuscript 'Development and evaluation of PlasmoPod: A cartridge-based nucleic acid amplification test for rapid malaria diagnosis and surveillance' has been provisionally accepted for publication in PLOS Global Public Health.

Best regards,

Sarah Auburn

Academic Editor